
# Virial expansion coefficients in the unitary Fermi gas

**Shimpei Endo**[1,2⋆]

**1** Frontier Research Institute for Interdisciplinary Sciences, Tohoku University,
6-3 Aoba, Aramaki, Aoba-ku, Sendai, Miyagi, 980-8578, Japan
**2** Department of Physics, Faculty of Science, Tohoku University,
6-3 Aoba, Aramaki, Aoba-ku, Sendai, Miyagi, 980-8578, Japan

⋆ shimpei.endo@nucl.phys.tohoku.ac.jp

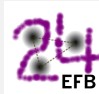 *Proceedings for the 24th edition of European Few Body Conference,*
## Abstract

Virial expansion is widely used in cold atoms to analyze high temperature strongly correlated many-body systems. As the n-th order virial expansion coefficient can be accurately obtained by exactly solving up to n-body problems, the virial expansion offers a few-body approach to study strongly correlated many-body problems. In particular, the virial expansion has successfully been applied to unitary Fermi gas. We review recent progress of the virial expansion studies in the unitary Fermi gas, in particular the fourth order virial coefficient.

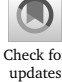

## 1 Introduction

Ultracold atoms have become nice platforms for studying various strongly correlated quantum phenomena thanks to their high controllability. In particular, the *s*-wave scattering length

between the atoms can be controlled by the Feshbach resonance [40, 42], so that systems with divergently large scattering length have been realized. Such systems are called unitary systems, and they have two important features: scale invariance and universality [46]. As a consequence, various few and many-body phenomena observed in cold atoms can be utilized as quantum simulators to understand similar phenomena found in other physical systems, such as nuclear and condensed matter systems [6, 30].

A prime example is the Efimov state [1, 37–39, 59]. As is shown by Efimov in 1970 [59], three-body systems interacting with divergently large $s$-wave scattering lengths show a series of weakly bound three-body states which can be related to each other by a discrete scale transformation. The Efimov states and their discrete scale invariance have been recently been observed in cold atom experiments in the unitary gases [33–36, 56]. The Efimov states have also been studied in various nuclear systems, such as halo states of neutron rich nuclei [31], Hoyle state of $^{12}$C [18, 19, 32], and triton [1, 59]. With recent studies on the Efimov states [20, 27, 29], it is now conjectured that the Efimov states and their characteristic length scale in various different atomic and nuclear systems may behave universally, and there have been active experimental and theoretical studies to scrutinize under what conditions the Efimov states can show such universality [10–16, 21, 22, 26, 28].

Another example of the universal quantum systems is the unitary Fermi gas [7, 43–45], a system of spin-up and spin-down fermions interacting with divergently large $s$-wave scattering length. Thanks to its scale invariance, the unitary Fermi gas in various physical systems should show the same behaviour independent of their energy scales and details of microscopic interactions [7, 62]. As the $s$-wave scattering length between the neutrons $a_{nn} \approx -23$ fm [8, 9] is so large that they can be approximately regarded as the unitary Fermi system, the knowledge of the equation of state of the unitary Fermi gas is relevant for understanding neutron stars [2, 3, 48–51]. For example, the low temperature equation of state of the unitary gas is useful for understanding the inner crust region of the neutron star [50–52]. The high temperature equation of state of the unitary gas is also relevant for understanding neutrino-neutron scattering processes during core-collapse supernovae and their neutrino radiation [2, 3]. By measuring the equation of state of the unitary Fermi gas in cold atom experiments, various universal physical quantities have been accurately determined, such as the Bertsch parameter [43–45] and Tan's contact [62, 81–84].

While these are few-body and many-body examples of universal phenomena, there is a way to bridge few-body and many-body phenomena. The virial expansion, sometimes also referred to as the cluster expansion, is a standard method in quantum statistical mechanics [41, 85]. A key idea of the virial expansion is that the quantum correlations between the particles generally becomes weaker as the temperature gets higher for a quantum many-body system. Therefore, we can take into account quantum correlations order by order in a systematic manner at high-temperature. Because the $n$-th order expansion coefficients of the virial expansion can be calculated from $n$-body solutions, the virial expansion provides a few-body perspective to study strongly correlated quantum many-body systems.

In this paper, we review how the virial expansion has been applied to the unitary Fermi gas. In particular, we compare the virial expansion coefficients of the equation of state of the unitary Fermi gas. While the virial expansion coefficients calculated from few-body solutions agree excellently up to the third order with those obtained from cold atom experiments, there is a discrepancy between them for the fourth order. We argue that the discrepancies are due to a non-monotonic behaviour of the equation of state of the unitary Fermi gas.

The paper is organized as follows: in the next section, we review the theoretical aspects of the virial expansion of the equation of state of the unitary Fermi gas. In Sec. 3, we discuss the fourth virial expansion coefficients of the unitary Fermi gas, comparing experimental and theoretical values. We argue that the origin of the discrepancies between them can be attributed

to the non-monotonic behaviour of the equation of state. We conclude in Sec. 4.

## 2 Virial expansion

In the virial expansinon, the thermodynamic potential $\Omega$ of the many-body system is expanded as [41, 85]

$$\frac{\Omega}{V} = -\frac{n_s k_B T}{\lambda^3} \left[ b_1 e^{\beta\mu} + b_2 e^{2\beta\mu} + b_3 e^{3\beta\mu} + b_4 e^{4\beta\mu} + ... \right], \tag{1}$$

where $k_B$ is the Boltzman constant, $\mu$ and $T = \beta^{-1}$ are the chemical potential and temperature, and $\lambda = \sqrt{2\pi\hbar/mk_B T}$ is the thermal de Brogie length. $n_s$ is the number of components: $n_s = 1$ for a spinless single component system, and $n_s = 2$ for a unitary Fermi gas which comprises of two spin states. The $n$-th order coefficients $b_n$ are called $n$-th order virial expansion coefficients, and they characterize the thermodynamic properties of the system. For example, pressure $P$, density $n$, and entropy $S$ of the system can be expressed using $b_n$ as

$$P = \frac{n_s k_B T}{\lambda^3} \left[ b_1 e^{\beta\mu} + b_2 e^{2\beta\mu} + b_3 e^{3\beta\mu} + b_4 e^{4\beta\mu} + ... \right], \tag{2}$$

$$n = \frac{n_s}{\lambda^3} \left[ b_1 e^{\beta\mu} + 2b_2 e^{2\beta\mu} + 3b_3 e^{3\beta\mu} + 4b_4 e^{4\beta\mu} + ... \right], \tag{3}$$

$$\begin{aligned}
\frac{S}{V} = &\frac{5n_s k_B}{2\lambda^3} \left[ b_1 e^{\beta\mu} + b_2 e^{2\beta\mu} + b_3 e^{3\beta\mu} + b_4 e^{4\beta\mu} + ... \right] \\
&- \frac{n_s \mu}{T\lambda^3} \left[ b_1 e^{\beta\mu} + 2b_2 e^{2\beta\mu} + 3b_3 e^{3\beta\mu} + 4b_4 e^{4\beta\mu} + ... \right] \\
&+ \frac{n_s k_B T}{\lambda^3} \left[ \frac{\partial b_1}{\partial T} e^{\beta\mu} + \frac{\partial b_2}{\partial T} e^{2\beta\mu} + \frac{\partial b_3}{\partial T} e^{3\beta\mu} + \frac{\partial b_4}{\partial T} e^{4\beta\mu} + ... \right].
\end{aligned} \tag{4}$$

At high temperature $T \to \infty$, the chemical potential becomes large and negative $\mu \to -\infty$, so that the fugacity $e^{\beta\mu}$ becomes very small. Therefore, the above virial expansion works well at high temperature. On the other hand, when the temperature becomes small and the system becomes quantum degenerate, the virial expansion breaks down. Indeed, this can be understood by noting that $e^{\beta\mu} \approx n\lambda^3$ at high temperature (see the first term of Eq. (3)). The system becomes quantum degenerate when $n\lambda^3 \sim 1$, which suggests $e^{\beta\mu} \sim 1$. Therefore, the virial expansion is valid if and only if the system is well away from quantum degeneracy.

The $n$-th order virial expansion coefficient can be calculated from up to n-body solutions. To see this, we represent the thermodynamic potential by the canonical partition function:

$$\begin{aligned}
\frac{\Omega}{V} = &-\frac{k_B T}{V} \log\left[ 1 + Z_1 e^{\beta\mu} + Z_2 e^{2\beta\mu} + Z_3 e^{3\beta\mu} + Z_4 e^{4\beta\mu} + ... \right] \\
= &-\frac{n_s k_B T}{\lambda^3} \left[ e^{\beta\mu} + \frac{Z_2 - \frac{1}{2}Z_1^2}{Z_1} e^{2\beta\mu} + \frac{Z_3 - Z_2 Z_1 + \frac{1}{3}Z_1^3}{Z_1} e^{3\beta\mu} \right. \\
&\left. + \frac{Z_4 - Z_3 Z_1 + Z_2 Z_1^2 - \frac{1}{2}Z_2^2 - \frac{1}{4}Z_1^4}{Z_1} e^{4\beta\mu} + ... \right],
\end{aligned} \tag{5}$$

where $Z_n$ is the canonical partition function of the $n$-body system. In the second line of Eq. (5), we have used $Z_1 = n_s V/\lambda^3$ for a uniform system. Comparing Eq. (1) and Eq. (5), we can see that $N$-th order virial coefficients can be calculated from $Z_1, Z_2, ... , Z_N$. As the canonical partition function of the $n$-body system $Z_n$ can be calculated by taking the canonical partition sum

of all the energy levels of the $n$-body system, this means that the $n$-th order virial coefficients can be accurately obtained if we know all the energy levels of the 1-body, 2-body,...., and $n$-body systems. Although it is practically impossible to obtain all the $n$-body energy eigenvalues for arbitrary $n$, it is still possible to accurately solve few-body problems for $n = 1, 2, 3, 4$ and obtain $b_n$ up to fourth order. One-body problem is trivially solved and we find $b_1 = 1$ (compare the first term of Eq. (1) and Eq. (5) ).

Even after accurately solving few-body problems and obtaining all the energy eigenvalues, we still need to take the canonical partition sum of them. They include not only the bound states, but also highly degenerate continuum states. This sum can be performed analytically for $n = 2$: the second virial coefficient is found to be related with the scattering phase shift by Beth and Uhlenbeck [80]

$$\frac{1}{\sqrt{2}}(b_2 - b_2^{(0)}) = \sum_i e^{-E_B^i/k_B T} + \frac{1}{\pi}\sum_\ell (2\ell + 1)\int_0^\infty dk \frac{d\delta_\ell}{dk} e^{-\frac{\lambda^2 k^2}{2\pi}} \qquad (6)$$

where $b_2^{(0)} = -1/4\sqrt{2}$ is the virial coefficient for the non-interacting system, $\delta_\ell$ is the $\ell$-th wave scattering phase shift, $E_B^i$ is the energy of the bound state. The first and second terms are bound-state and continuum state contributions, respectively. Using $\delta_{\ell>0} = 0$, $k\cot\delta_0 = -1/a$, and that there is no bound state for the unitary Fermi gas, one finds $b_2 = 3/4\sqrt{2}$. This is found to be consistent with the unitary Fermi gas experiment [43–45].

On the other hand, for a system with more than two particles, it is not easy to take the canonical partition sum of such highly degenerate energy levels analytically. We therefore often consider a finite size $n$-body system under confinement: after calculating $b_n$ for such a trapped system, we take the infinite system limit to obtain $b_n$ in a homogeneous system. We typically consider a system in an isotropic harmonic trap because the unitary Fermi gas in an isotropic harmonic trap possesses good symmetry feature. In particular, hyper-radial and hyper-angular equations become separable for a unitary Fermi gas since the system is scale invariant [7]. This facilitates solving few-body problem in this system both numerically and analytically. Using analytical solutions of the unitary three-body system in the harmonic trap [78], Liu, Hu, and Drummond have accurately obtained $b_3$ of the unitary Fermi gas to be $b_3 = -0.29095295$ [79] after extrapolating it to a homogeneous system. This is consistent with what is experimentally obtained in the unitary Fermi gas experiment $b_3 = -0.29(2)$ [43, 44]. Other theory groups have independently calculated $b_3$ and obtained essentially the same value: Castin and Endo [23] used a contour integral in the complex plane to take the canonical partition sum of the energy levels and obtained $b_3 = -0.2909529965421$. Rakshit, Daily, and Blume [76] used numerical few-body solutions of the harmonically trapped system, and obtained $b_3 = -0.2909529965797$ after extrapolating it to a homogeneous system. We note that some others used field theoretic methods to directly calculate $b_3$ without using the trapped system: Kaplan and Sun obtained $b_3 = -0.2931$ [64] and Leyronas obtained $b_3 = -0.2909$ [75]

## 3 Fourth order virial expansion coefficient in the unitary Fermi gas

From the equation of state of the unitary Fermi gas experiment in cold atoms, the fourth order virial expansion coefficient has also been observed [43, 44]. By using $b_1$, $b_2$, $b_3$ values obtained theoretically as above, they have fitted $b_4$ to their observed equation of state at high temperature. In 2010-2012, two experimental groups have found $b_4 = 0.065(10)$ [44] and $b_4 = 0.065(15)$ [43], respectively. At the time, there was one work which theoretically calculated the fourth order virial coefficient $b_4$ [76]. However, its value $b_4 = -0.047(4)$ was in contradiction with what was found in the experiments.

To solve this puzzle, several groups have independently calculated $b_4$ of the unitary Fermi gas. We compare their results in Table 1. Ngampruetikorn, Parish, and Levinsen [77] used a field theoretic method for a homogeneous system and approximately obtained $b_4 \approx 0.03$. Endo and Castin [25] applied the contour integral method to take the canonical partition sum in a homogeneous system and obtained $b_4 = 0.031(1)$. While their contour integral method uses exact few-body solutions and has a rather small error bar, we note that it is based on a conjecture that $b_4$ can be obtained from a similar contour integral formula as $b_3$ supplemented with a counter term they introduce to cancel the dimer-dimer like contribution in the four-body equation kernel [23, 25]. Yan and Blume [77] use the Path Integral Monte Carlo method to obtain $b_4$ in a harmonic trap, and obtain $b_4 = 0.047(18)$ after extrapolating it to a homogeneous system by taking $\omega \to 0$. Therefore, all the theoretical results [24, 25, 77] obtained after 2013 seem to suggest $b_4 = 0.03$. This value is more consistent with the experimental results $b_4 = 0.065$ than that in Ref. [76], but in a slight disagreement.

The origin of this disagreement can be attributed to a peculiar non-monotonic behaviour of the equation of state of the unitary Fermi gas. In Refs [24, 25], it is found that the fourth virial expansion coefficient of the harmonically trapped system $B_4(\beta\hbar\omega)$ shows a non-monotonic behaviour as a function of $\beta\hbar\omega$: $B_4(\beta\hbar\omega)$ is monotonically increasing (decreasing) for $\beta\hbar\omega \gtrsim 1.5$ ($\beta\hbar\omega \lesssim 1.5$). This nonmonotonicity can cause a problem when extrapolating $B_4(\beta\hbar\omega)$ to a homogeneous system $b_4$ by taking $\omega \to 0$ limit. Indeed, in Ref. [76], $B_4(\beta\hbar\omega)$ has been accurately calculated for $\beta\hbar\omega \gtrsim 1.5$, and it is extrapolated to $\omega \to 0$ limit without noticing this non-monotonic behaviour. This is why $b_4 = -0.047(4)$ obtained in Ref. [76] is very different from the other results. In Refs [24, 25], $B_4(\beta\hbar\omega)$ has been accurately calculated for wider range of $\beta\hbar\omega$, including $\beta\hbar\omega \lesssim 1.5$ region. Their results for $B_4(\beta\hbar\omega)$ is in excellent agreement with that obtained in Ref. [76] for $\beta\hbar\omega \gtrsim 1.5$. With the $B_4(\beta\hbar\omega)$ results in $\beta\hbar\omega \lesssim 1.5$ region, Ref [24] have more accurate extrapolation to $\omega \to 0$ limit taking the nonmonotonicity into account, and obtained $b_4 = 0.047(18)$. Ref [25] has a much smaller error bar $b_4 = 0.031(1)$ firstly because the limit $\omega \to 0$ is taken analytically and does not suffer from the extrapolation error, and secondly because the canonical partition sum of all the energy levels is done by the contour integral method which does not suffer from an energy cutoff. We also note that Ref. [77] calculated $b_4$ for a homogeneous system without using the harmonic trap, and therefore does not suffer from such an extrapolation subtlety.

The physical origin of the nonmonotonicity can be understood by noting that the fourth virial coefficient mainly comprises of two non-trivial contributions

$$B_4 - B_4^{(0)} = B_{\uparrow\uparrow\uparrow\downarrow} + \frac{1}{2}B_{\uparrow\uparrow\downarrow\downarrow}, \tag{7}$$

$B_4^{(0)}$ is the fourth virial coefficient for a non-interacting system and trivially gives a constant shift in the uniform limit $B_4^{(0)}(\beta\hbar\omega = 0) = -1/256$. $B_{\uparrow\uparrow\uparrow\downarrow}$ and $B_{\uparrow\uparrow\downarrow\downarrow}$ are the fourth virial coefficients of a 3 spin-up + 1 spin-down system, and a 2 spin-up + 2 spin-down system, respectively (we note that 3 spin-up + 1 spin-down system is equivalent to 1 spin-up + 3 spin-down system). $B_{\uparrow\uparrow\uparrow\downarrow}$ is positive and monotonically decreasing, while $B_{\uparrow\uparrow\downarrow\downarrow}$ is negative and monotonically increasing [24, 25]. This difference can be attributed to stronger Pauli exclusion effect in a 3 spin-up + 1 spin-down system compared with a 2 spin-up + 2 spin-down system. As $B_4$ is a sum of monotonically increasing and decreasing terms, the resulting $B_4$ shows the nonmonotonicity: the effect of $B_{\uparrow\uparrow\downarrow\downarrow}$ is larger for $\beta\hbar\omega \gtrsim 1.5$ because the 2 spin-up + 2 spin-down system has smaller energy than the 3 spin-up + 1 spin-down system and therefore has a larger contribution in the strong trap limit $\beta\hbar\omega \to \infty$. Conversely, in the $\beta\hbar\omega \ll 1$ regime, the Pauli exclusion effect plays less significant role, and crudely speaking we can regard $B_{\uparrow\uparrow\uparrow\downarrow} \sim B_{\uparrow\uparrow\downarrow\downarrow}$. Since $B_{\uparrow\uparrow\uparrow\downarrow}$ has a twice contribution, originating from 3 spin-up + 1 spin-down and 3 spin-down + 1 spin-up systems), compared with $B_{\uparrow\uparrow\downarrow\downarrow}$, $B_{\uparrow\uparrow\uparrow\downarrow}$ plays more dominant role in the $\beta\hbar\omega \ll 1$ regime.

Table 1: Comparison of $b_4$ obtained in experiments (first and second rows) and theories (third to sixth rows).

| Method | $b_4$ | Reference |
|---|---|---|
| Cold atom experiment by ENS | 0.065(15) | [44] |
| Cold atom experiment by MIT | 0.065(10) | [43] |
| Numerical 4-body solution in a harmonic trap + extrapolation $\omega \to 0$ | -0.047(4) | [76] |
| Feynman diagram (homogeneous system) | $\approx 0.03$ | [77] |
| Contour integral method (homogeneous system) | 0.031(1) | [25] |
| Path Integral Monte Carlo in a harmonic trap + extrapolation $\omega \to 0$ | 0.047(18) | [24] |

The nonmonotonicity also explains the discrepancy between the experimental values $b_4 = 0.065$ [43, 44] and theoretical values $b_4 = 0.03$ [24, 25, 77]. In Ref. [5], the equation of state of the homogeneous unitary Fermi gas has been accurately calculated by the diagrammatic Monte Carlo calculation. The authors have found that the equation of state as a function of $e^{\beta\mu}$ shows a non-monotonic behaviour at $e^{\beta\mu} \sim 1$, in a similar manner as explained above for a harmonically trapped system. They have found that their equation of state at $e^{\beta\mu} \sim 1$ is consistent with the theory value $b_4 = 0.031(1)$ [25], while it is also consistent with the experimental equation of state for $e^{\beta\mu} \gtrsim 1$. It turns out that the experimental values $b_4 = 0.065$ reported in Refs [43, 44] are plagued by the nonmonotonicity: by extrapolating the equation of state to $e^{\beta\mu} \to 0$ limit in order to obtain $b_4$ with the experimental data mostly in $e^{\beta\mu} \gtrsim 1$ region, they seem to have overestimated $b_4$. Therefore, it seems that $b_4 = 0.031(1)$, which comprises of $b_4 = b_{\uparrow\uparrow\uparrow\downarrow} + \frac{1}{2}b_{\uparrow\uparrow\downarrow\downarrow} - \frac{1}{32}$ with $b_{\uparrow\uparrow\uparrow\downarrow} = 0.1838(4)$ and $b_{\uparrow\uparrow\downarrow\downarrow} = -0.244(2)$ [25], is currently the most plausible value. Furthermore, from the behaviour of the equation of state at $e^{\beta\mu} \lesssim 1$, it is likely that the fifth and sixth order virial coefficients should be $b_5 > 0$ and $b_6 < 0$, although further studies are required to accurately determine the values of $b_5$ and $b_6$.

The nonmonotonicity in the homogeneous system as a function of $e^{\beta\mu}$ is closely related with that in the harmonically trapped system as a function of $\beta\hbar\omega$. This can be understood by assuming that the effect of the strongly correlated many-body medium can be, crudely speaking, regarded as providing an effective trapping potential to particles embedded in the medium. As the energy scale of the unitary Fermi gas is given universally by $T_F$, the Fermi temperature, we can regard this effective trapping potential to be $\hbar\omega \sim k_B T_F$. In this way, we can find that the crossover point of the nonmonotonicity for the homogeneous system $e^{\beta\mu} \approx n\lambda^3 \approx (T_F/T)^{3/2} \sim 1$ agrees with that in the harmonically trapped system $\beta\hbar\omega \sim T_F/T \sim 1$.

## 4 Conclusion

In this paper, we review the virial expansion of the equation of state of the unitary Fermi. The unitary Fermi gas has been realized in cold atom experiments, and its universal equation of state has been observed. Its high temperature behaviour is characterized by the virial expansion coefficients $b_n$. As $b_n$ can be accurately calculated from exact few-body solutions, the virial expansion offers a few-body perspective to study strongly correlated many-body systems. We review how $b_n$ for $n \le 4$ has been theoretically obtained in the unitary Fermi gas, and compared with those observed experimentally. While the theoretical [23, 64, 75, 76, 79] and experimental [43, 44] values agree excellently for $n \le 3$, there is a discrepancy for $b_4$. The discrepancy originates from the non-monotonic behaviour of the equation of state, which has plagued the extrapolation procedure to extract $b_4$ from the experimental data. We therefore conclude that the theoretically reported value $b_4 = 0.031(1)$ [24, 25, 77] seems to be the most plausible value, rather than the experimental one $b_4 = 0.065(15)$ [43, 44].

Accurate determination of the high-temperature equation of state of the unitary Fermi gas is useful for understanding the neutron star and supernova physics. In Ref. [2], for example, it is argued that there exists a region in the neutron star which can be regarded as a high-temperature unitary Fermi system of neutrons. It is shown that the neutrino-neutron scattering cross section, relevant quantity for understanding core-collapse supernovae and their neutrino radiation, can be obtained from the high-temperature equation of state of the unitary Fermi gas, and the virial expansion of the unitary Fermi gas is applied.

The virial expansion is also useful in studying universal thermodynamic behaviour of other strongly correlated systems. In particular, it is a powerful tool in investigating many-body systems in the presence of Efimov trimers. Such systems have attracted a lot of interests since the realization of the so-called unitary Bose gas system [65,71–73], in which the Efimov trimers can play relevant role in the strongly correlated many-body system [74]. The virial expansion enables accurately incorporating the trimer degree of freedom into the many-body theory. The third virial expansion coefficient has been accurately calculated for the unitary Bose gas taking into account the Efimov trimers [61]. The third virial coefficients in the presence of Efimov effects have also been accurately calculated for a mass imbalanced two-component Fermi system [47] and for a mass imbalanced two-component Bose system [53]. We expect that other universal clusters, such as universal trimers found by Kartavtsev and Malykh [58, 66–68, 96] and the tetramers [69, 70], can be incorporated into the many-body calculation with the virial expansion.

# Acknowledgements

S. Endo acknowledges support from JSPS KAKENHI Grant-in-Aid for Research Activity Start-up 19K21028.

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
