# Peer review of "Virial expansion coefficients in the unitary Fermi gas"

_SciPost Physics Proceedings, doi:SciPost Phys. Proc. 3, 049 (2020)_

## Round 1 · Referee Report · Anonymous (Referee 1) · 2019-12-12

Report

The author reviews the study of virial expansion in a unitary Fermi gas.
The general background of the virial expansion is clearly explained in the connection with few-body and many-body physics.

I recommend the publication of this article after addressing following issues.

1) In the introduction, the author mentions that a unitary Fermi gas is relevant for understanding neutron stars.
I think that more detailed explanation with some references is required because the system scales are quite different between these two systems.
While the scattering length in cold atoms is generally the order of nano(or micro)-meters, the neutron-neutron scattering length is -18.5 femto-meters.

2) Since the virial expansion itself cannot directly be applied to neutron star physics where the temperature is quite low, I suggest the author additionally refer to the supernova matter at finite temperature.
In Phys. Rev. C 96, 055804 (2017), a neutrino-nucleon scattering properties have been derived from the 4th virial expansion of a unitary Fermi gas.

3) I think that Eqs. (2)-(4) for the virial equation of state should explicitly involve the 4-th order coefficient b_4 since in the latter part the author reviews b_4.

---

## Round 2 · Author Response

We appreciate the referee for carefully reading my manuscript and appreciating it positively. All the 3 major comments (listed below) raised by the referee are very helpful, and I have improved the manuscript following the comments (see below). In addition, I have also revised some minor typos and mistakes (see List of changes):
>1) In the introduction, the author mentions that a unitary Fermi gas is relevant for understanding neutron stars.
>I think that more detailed explanation with some references is required because the system scales are quite different between these ?>two systems.While the scattering length in cold atoms is generally the order of nano(or micro)-meters, the neutron-neutron >scattering length is -18.5 femto-meters.
I have added some relevant references which have studied the relation between the neutron star and cold atoms. I have also added the following sentences in the 3rd paragraph of the Introduction to make it easier for the readers to understand why the two physical systems with very different energy scales should behave in the same manner: "Thanks to its scale invariance, the unitary Fermi gas in various physical systems should show the same behaviour independent of their energy scales and details of microscopic interactions. As the $s$-wave scattering length between the neutrons $a_{nn}\approx -23 $ fm is so large that they can be approximately regarded as the unitary Fermi system, the knowledge of the equation of state of the unitary Fermi gas is relevant for understanding neutron stars."
>2) Since the virial expansion itself cannot directly be applied to neutron star physics where the temperature is quite low, I suggest >the author additionally refer to the supernova matter at finite temperature. In Phys. Rev. C 96, 055804 (2017), a neutrino-nucleon >scattering properties have been derived from the 4th virial expansion of a unitary Fermi gas.
The paper pointed out by the referee is very relevant for this manuscript, and I appreciate the referee for pointed this out. I have cited it, and added the following sentences in the Introduction and Conclusions, so that the readers can more easily understand why the high-temperature EOS is relevant for neutron star physics:
Introduction: "For example, the low temperature equation of state of the unitary gas is useful for understanding the inner crust region of the neutron star [43,44,47]. The high temperature equation of state of the unitary gas is also relevant for understanding neutrino-neutron scattering processes during core-collapse supernovae and their neutrino radiation [45,46]".
Conclusion: "Accurate determination of the high-temperature equation of state of the unitary Fermi gas is useful for understanding the neutron star and supernova physics. In Ref.~[45], for example, it is argued that there exists a region in the neutron star which can be regarded as a high-temperature unitary Fermi system of neutrons. It is shown that the neutrino-neutron scattering cross section, relevant quantity for understanding core-collapse supernovae and their neutrino radiation, can be obtained from the high-temperature equation of state of the unitary Fermi gas. "
>3) I think that Eqs. (2)-(4) for the virial equation of state should explicitly involve the 4-th order coefficient b_4 since in the latter part the author reviews b_4.
I have added the fourth order terms in all the equations.
>1) In the introduction, the author mentions that a unitary Fermi gas is relevant for understanding neutron stars.
>I think that more detailed explanation with some references is required because the system scales are quite different between these ?>two systems.While the scattering length in cold atoms is generally the order of nano(or micro)-meters, the neutron-neutron >scattering length is -18.5 femto-meters.
I have added some relevant references which have studied the relation between the neutron star and cold atoms. I have also added the following sentences in the 3rd paragraph of the Introduction to make it easier for the readers to understand why the two physical systems with very different energy scales should behave in the same manner: "Thanks to its scale invariance, the unitary Fermi gas in various physical systems should show the same behaviour independent of their energy scales and details of microscopic interactions. As the $s$-wave scattering length between the neutrons $a_{nn}\approx -23 $ fm is so large that they can be approximately regarded as the unitary Fermi system, the knowledge of the equation of state of the unitary Fermi gas is relevant for understanding neutron stars."
>2) Since the virial expansion itself cannot directly be applied to neutron star physics where the temperature is quite low, I suggest >the author additionally refer to the supernova matter at finite temperature. In Phys. Rev. C 96, 055804 (2017), a neutrino-nucleon >scattering properties have been derived from the 4th virial expansion of a unitary Fermi gas.
The paper pointed out by the referee is very relevant for this manuscript, and I appreciate the referee for pointed this out. I have cited it, and added the following sentences in the Introduction and Conclusions, so that the readers can more easily understand why the high-temperature EOS is relevant for neutron star physics:
Introduction: "For example, the low temperature equation of state of the unitary gas is useful for understanding the inner crust region of the neutron star [43,44,47]. The high temperature equation of state of the unitary gas is also relevant for understanding neutrino-neutron scattering processes during core-collapse supernovae and their neutrino radiation [45,46]".
Conclusion: "Accurate determination of the high-temperature equation of state of the unitary Fermi gas is useful for understanding the neutron star and supernova physics. In Ref.~[45], for example, it is argued that there exists a region in the neutron star which can be regarded as a high-temperature unitary Fermi system of neutrons. It is shown that the neutrino-neutron scattering cross section, relevant quantity for understanding core-collapse supernovae and their neutrino radiation, can be obtained from the high-temperature equation of state of the unitary Fermi gas. "
>3) I think that Eqs. (2)-(4) for the virial equation of state should explicitly involve the 4-th order coefficient b_4 since in the latter part the author reviews b_4.
I have added the fourth order terms in all the equations.

---

## Round 2 · List of Changes

(i) I have changed the third paragraph of the Introduction, following the comments 1) and 2) of the referee . In short, I have added more discussions and references explaining why the unitary cold atom sysetm is relevant for understanding the neutron star.
(ii) In Eq. (1)-(5), I have added the fourth order terms with b_4.
(iii) There is mistake in Eq (7): a non-interacting contribution B_4^{(0)} is missing. I have revised it and the subsequent sentence accordingly.
(iv) In the fifth paragraph of Sec. 3, I have added the values of b_{up up up down} and b_{up up down down}. This is because these spin-dependent quantities are necessary for calculating the neutrino-neutron scattering cross section in Phys. Rev. C 96, 055804 (2017) pointed out by the referee's comment, and I think some readers should be interested in knowing the values of b_{up up up down} and b_{up up down down}, in addition to b_4.
(v) The second paragraph of the Conclusion is revised following the comments 1) and 2) of the referee. In short, it explains how the high-temperature equation of state obtained from the virial coefficients of the unitary Fermi gas can be directly applied to neutron-star physics.
(vi) Some minor English typos are revised.
(ii) In Eq. (1)-(5), I have added the fourth order terms with b_4.
(iii) There is mistake in Eq (7): a non-interacting contribution B_4^{(0)} is missing. I have revised it and the subsequent sentence accordingly.
(iv) In the fifth paragraph of Sec. 3, I have added the values of b_{up up up down} and b_{up up down down}. This is because these spin-dependent quantities are necessary for calculating the neutrino-neutron scattering cross section in Phys. Rev. C 96, 055804 (2017) pointed out by the referee's comment, and I think some readers should be interested in knowing the values of b_{up up up down} and b_{up up down down}, in addition to b_4.
(v) The second paragraph of the Conclusion is revised following the comments 1) and 2) of the referee. In short, it explains how the high-temperature equation of state obtained from the virial coefficients of the unitary Fermi gas can be directly applied to neutron-star physics.
(vi) Some minor English typos are revised.

---

## Editorial Decision

published